# Pulsed low-energy stimulation initiates electric turbulence in cardiac tissue

**Rupamanjari Majumder**[1]*, **Sayedeh Hussaini**[1,2], **Vladimir S. Zykov**[1], **Stefan Luther**[1,2], **Eberhard Bodenschatz**[1,3]

**1** Max Planck Institute for Dynamics and Self-Organization, Göttingen, Germany, **2** Institute for Dynamics of Complex Systems, University of Göttingen, Göttingen, Germany, **3** Laboratory of Atomic and Solid-State Physics and Sibley School of Mechanical and Aerospace Engineering, Cornell University, Ithaca, New York, United States of America

* rupamanjari.majumder@ds.mpg.de

**Data Availability Statement:** All relevant data are within the manuscript and its Supporting information files.

**Funding:** This work was supported by the German Center Cardiovascular Research (DZHK) to EB

## Abstract

Interruptions in nonlinear wave propagation, commonly referred to as wave breaks, are typical of many complex excitable systems. In the heart they lead to lethal rhythm disorders, the so-called arrhythmias, which are one of the main causes of sudden death in the industrialized world. Progress in the treatment and therapy of cardiac arrhythmias requires a detailed understanding of the triggers and dynamics of these wave breaks. In particular, two very important questions are: 1) What determines the potential of a wave break to initiate re-entry? and 2) How do these breaks evolve such that the system is able to maintain spatiotemporally chaotic electrical activity? Here we approach these questions numerically using optogenetics in an *in silico* model of human atrial tissue that has undergone chronic atrial fibrillation (cAF) remodelling. In the lesser studied sub-threshold illumination régime, we discover a new mechanism of wave break initiation in cardiac tissue that occurs for gentle slopes of the restitution characteristics. This mechanism involves the creation of conduction blocks through a combination of wavefront-waveback interaction, reshaping of the wave profile and heterogeneous recovery from the excitation of the spatially extended medium, leading to the creation of re-excitable windows for sustained re-entry. This finding is an important contribution to cardiac arrhythmia research as it identifies scenarios in which low-energy perturbations to cardiac rhythm can be potentially life-threatening.

## Author summary

Electric turbulence in the heart is associated with complex spatiotemporal dynamics of nonlinear excitation waves. This life-threatening state is initiated by wavebreaks and maintained by self-organized vortices of abnormal excitation. Control over this state is achieved via defibrillation, a method whose efficacy relies on understanding the role of electric vortices in the onset and perpetuation of the turbulent state. However, in biological tissue, these vortices remain largely elusive in experiments due to limitations of visualisation. In order to study the spatiotemporal evolution of vortices in the heart precisely, reversibly and in real-time, we apply computational cardiac optogenetics on a

("Allianz für die Regeneration bei Herzinsuffizienz", Grant no 81Z0300404) and to SL (Project MD 28 grant no. 81Z0300403/81Z0300114) and by the German Research Foundation (Research Centre SFB 1002, Project C3) to SL. The funders had no role in study design, data collection and analysis, decision to publish, or preparation of the manuscript.

**Competing interests:** The authors have declared that no competing interests exist.

mathematical model for human atrial tissue. Our study demonstrates a heretofore unidentified mechanism of wavebreak initiation, in the absence of standard markers of excitable medium vulnerability. This finding is an important contribution to cardiac arrhythmia research as it identifies how low-energy perturbations to cardiac rhythm can be potentially life-threatening, and render a defibrillation strategy to backfire.

## Introduction

Spiral waves occur as short- or long-lived transients in various natural excitable systems [1–8]. In the heart, they occur as abnormal electrical waves underlying lethal cardiac arrhythmias (tachycardia and fibrillation), which are the most common precursors of sudden cardiac death (SCD) [9]. The spontaneous breakup of electrical spiral waves in cardiac tissue leads to a state of electric turbulence in the heart, which is clinically recognized as fibrillation. Control of this turbulent state is essential for restoration of normal cardiac function and prevention of SCD. The state of the art clinical approach to exert such control, is a technique called defibrillation. Conventional defibrillation involves applying a large-amplitude electric field to the heart to achieve organ-wide electrical synchronization [10]. However, the therapy is extremely painful, traumatic, and produces harmful side effects that over the years have led to increased demand for low-energy alternatives, particularly ones that involve the use of pulse trains as opposed to the continuous application of shocks. The effective design and development of novel therapeutic strategies to control electric turbulence in the heart requires a detailed understanding of the spatiotemporal evolution of this turbulent state: its triggers, its development, and its dependence on the nature of the underlying substrate.

In complex excitable systems, such as cardiac tissue, the dynamics of nonlinear spiral waves is determined primarily by the excitability of the medium. Hence, the modulation of excitability by, for example, varying its amplitude, frequency, degree of synchronization, and spatial scale, phenomena such as drift [11, 12], deformation [13], block [13], meander [12, 14], breakup [13, 15], and suppression of spiral waves [16] can be observed. Of these phenomena, breakup essentially results in electric turbulence in the heart and is mediated by the formation of wavebreaks close to or away from the core of intact spirals. Therefore, understanding the origin of wavebreaks is an important first step in theorizing the evolution of the turbulent state, which is essential for developing an appropriate scheme for its control. Such in-depth investigations require powerful research tools to exercise precise spatiotemporal control over the properties of the system down to the fundamental, i.e., cellular, level. Unfortunately, such experimental tools that allow reversible control of excitability in real time are hard to find. Therefore, one resorts to the use of numerical techniques to address these fundamental research questions in computer models of cardiac tissue.

Recently, optogenetics has offered a plausible solution to overcome the challenge of controlling excitability in experiments with cardiac tissue [17]. This technique relies on genetically engineering light sensitivity into cardiac muscle cells to optical allow control over the direction and flow of current across cell membranes [18]. Thus, using light of specific wavelengths, intensity, duration and projection patterns, one can control wave propagation in cardiac tissue [19–21]. Moreover, *ex vivo* studies conducted in small mammalian hearts show that optogenetics may provide a viable alternative to conventional defibrillation [22–25]. However, the mechanisms underlying success or failure of the method remain incompletely understood, because of the general lack of knowledge regarding the properties and effects of optogenetics in the heart, at nonstimulating (sub-threshold) light intensities [21, 26]. Although it is widely-

accepted that stimulating (supra-threshold) disturbances, be it of electrical [27], or optical [19] nature, can induce breakup of spiral waves, and thereby promote electric turbulence in cardiac tissue, the corresponding response at sub-threshold stimulation remains to be examined. The closest study in this regard was performed by Hussaini et al. [28], who show that spatial modulation of excitability in the sub-threshold stimulation régime can induce drift of a spiral wave in cardiac tissue. However, in so far as literature is concerned, sub-threshold stimulation has never been studied in the context of wavebreak in cardiac tissue. In this study we demonstrate *in silico* a heretofore unknown mechanism of wavebreak initiation by sub-threshold stimulation of optogenetically modified human atrial tissue. For simplicity, we use a two dimensional (2D) simulation domain, which represents a sheet of tissue inside the atrial wall. We observe that global periodic perturbations with high intensity sub-threshold optical stimuli, lead to the development of wavebreaks by "conditioning" of the wavelength. Such conditioning involves modulating the distribution of excitation along an arm of the intact spiral, which prompts the creation of excitation windows for wave break and re-entry. Because this observation describes a scenario in which a plausible low-energy control scheme (applied to the surface of the heart) backfires and promotes fibrillation, we believe that our findings are of central importance in the design and development of defibrillation strategies for use in medical devices.

## Materials and methods

Electrical activity in cardiac tissue was modeled using the following reaction-diffusion-type equation:

$$\frac{dV}{dt} = \nabla.\mathcal{D}\nabla V - \frac{I_{ion}}{C_m} \tag{1}$$

where $V$ represents the transmembrane potential (in mV) developed across single cardiomyocytes, $C_m$ is the specific capacitance (in $\mu$F/cm$^2$) of the cell membrane, $\mathcal{D}$ represents the diffusion coefficient for intercellular coupling, and $I_{ion}$ represents the total ionic current produced by each individual cell. For human atrial tissue, $I_{ion}$ was formulated according to the Courtemanche-Ramirez-Nattel (CRN) model [29]. It is a sum of 12 ionic currents: fast $Na^+$ ($I_{Na}$), inward rectifier $K^+$ ($I_{K1}$), transient outward $K^+$ ($I_{to}$), ultra-rapid $K^+$ ($I_{Kur}$), rapid and slow delayed rectifier $K^+$ ($I_{Kr}$ and $I_{Ks}$, respectively), $Na^+$ and $Ca^{2+}$ background ($I_{BNa}$, and $I_{CaL}$), $Na^+$/$K^+$ pump, $Ca^{2+}$ pump ($I_{pCa}$), $Na^+/Ca^{2+}$ exchanger ($I_{NaCa}$) and $L-type$ $Ca^{2+}$ current ($I_{CaL}$). The parameters of the CRN model were adjusted to reproduce the action potential of the normal atrial working myocardium [30]. In particular, the maximum conductance for $I_{K1}$, i.e., $G_{K1}$ was changed from 0.09 nS/pF in the original model, to 0.117 nS/pF This resulted in an APD$_{90}$ value of 284 ms. Here APD$_{90}$ refers to the amount of time during an action potential, when the membrane voltage is less than 90% repolarised. In general, APD$_X$ is calculated as $T_1 - T_0$, where $T_1$ and $T_0$ are the time instants at which the membrane voltage crosses the threshold value $V_{th}$. $V_{th}$ is calculated according to Eq 2

$$V_{th} = V_{max} - \frac{X}{100}(V_{max} - V_{min}) \tag{2}$$

$V_{max}$ and $V_{min}$ represent, respectively, the maximum and minimum values of the membrane voltage, as recorded during an action potential, and $X$ represents the degree of repolarization of the cell membrane, that we are interested in looking at. A choice of $\mathcal{D} = 0.0023$ cm$^2$/ms, produced a conduction velocity (CV) of 69.75 cm/s in the two-dimensional healthy tissue domain. In order to model the action potential during chronic atrial fibrillation (AF) remodelling, the maximal conductances of $I_{to}$, and $I_{CaL}$ were reduced by 85%, and 74%, respectively,

$G_{K1}$ was increased by 250%, the time constant for activation of $I_{CaL}$ was increased by 62%, the activation curves for $I_{to}$ and $I_{CaL}$ wer shifted by +16 mV and -5.4 mV, respectively, while the inactivation curve for $I_{Na}$ was shifted by +1.6 mV [31–33]. These parameter adjustments reduced the wavelength (at 1Hz electrical pacing) from ≃18 cm in healthy tissue, to ≃5 cm, allowing us to fit the spiral into a smaller simulation domain ($512 \times 512$, as opposed to $2048 \times 2048$) without causing it to break up. This improved the computational cost-effectiveness by a factor of 20. The spiral wave in our study meandered with a hypocycloidal tip trajectory, and survived for longer than 10s of simulation time. For simulations with the CRN model, we used a time step $\delta t = 0.02$ ms and a grid spacing $\delta x = \delta y = 0.022$ cm.

For the neonatal mouse ventricular model, $I_{ion}$ was expressed as a sum of 16 ionic currents, according to the Wang and Sobie [34]. These include, the fast $Na^+$ current ($I_{Na}$), the background $Na^+$ and $Ca^{2+}$ currents ($I_{Nab}$ and $I_{Cab}$), the L-type and T-type $Ca^{2+}$ currents ($I_{CaL}$ and $I_{CaT}$), the $Ca^{2+}$ pump current ($I_{pCa}$), the $Na^+/Ca^{2+}$ exchanger ($I_{NaCa}$), $Na^+/K^+$ pump ($I_{NaK}$), slow and fast components of the transient outward $K^+$ currents ($I_{Kto,f}$ and $I_{Kto,s}$), slow and rapid delayed rectifier $K^+$ currents ($I_{Ks}$ and $I_{Kr}$), ultrarapid delayed rectifier $K^+$ current ($I_{Kur}$), sustained outward $K^+$ current ($I_{Kss}$), inward rectifier ($I_{K1}$), and the $Ca^{2+}$-activated $Cl^-$ current ($I_{Cl,Ca}$). We used $\mathcal{D} = 0.00095$ cm/ms, which led to a CV of 43.9 cm/s. The 2D simnulation domain contained $200 \times 200$ grid points.

In order to incorporate the effects of optogenetics, our models were combined with a 4-state model for voltage- and light-sensitive Channelrhodopsin-2 [35] with modifications.

## The optogenetic model

Our Optogenetic model consists of a hybrid formulation, based on the models by [35], and [36] as described in Eqs 3–17. Note that, unlike the model by [36], we incorporated voltage sensitivity in the conductance of the ChR2 ion channel, but left it out in case of other kinetic parameters, which are considered voltage sensitive by [35]. The basic idea behind implementing this formulation was to find a compromise between obtaining a wide range of parameters for the sub-threshold activity and using light intensities typically used in experiments. We found that neither the Boyle model, nor the Williams model could be used in their direct form for this purpose. Hence the hybrid setup.

$$\frac{dV}{dt} = \mathcal{D}\nabla^2 V - \frac{I_{ion} + I_{ChR2}}{C_m}. \tag{3}$$

$$I_{ChR2} = g_{ChR2}G(V)(O_1 + \gamma O_2)(V - E_{ChR2})S_{cell}C_m, \tag{4}$$

$$\frac{dC_1}{dt} = G_r C_2 + G_{d1} O_1 - k_1 C_1, \tag{5}$$

$$\frac{dC_2}{dt} = G_{d2} O_2 - (k_2 + G_r)C_2, \tag{6}$$

$$\frac{dO_1}{dt} = k_1 C_1 - (G_{d1} + e_{12})O_1 + e_{21}O_2, \tag{7}$$

$$\frac{dO_2}{dt} = k_2 C_2 - (G_{d2} + e_{21})O_2 + e_{12}O_1, \tag{8}$$

$$O_1 + O_2 + C_1 + C_2 = 1, \tag{9}$$

$$G(V) = 10.6408 - \frac{14.6408 exp(-V/42.7671)}{V}, \tag{10}$$

Here $O_1$, $O_2$, $C_1$ and $C_2$ represent the open and closed states of the ChR2 ion channel, $S_{cell}$ is the cell surface area, $k_1$, $k_2$, $G_{d1} = 0.1$, $G_{d2} = 0.05$, $G_r = 0.004$, $e_{12}$, and $e_{21}$ are the kinetic parameters, described as follows:

$$k_1 = 0.8535 \mathcal{F} p, \tag{11}$$

$$k_2 = 0.14 \mathcal{F} p, \tag{12}$$

$$\mathcal{F} = (6 \times 10^{-5} E_e \lambda w_{loss}), \tag{13}$$

$$\frac{\partial p}{\partial t} = \frac{S_0 - p}{\tau_{ChR2}}, \tag{14}$$

$$S_0 = 0.5(1 + tanh(120(E_e - 0.1))), \tag{15}$$

$$e_{12} = 0.011 + 0.005 log(E_e/0.024), \tag{16}$$

$$e_{21} = 0.008 + 0.004 log(E_e/0.024) \tag{17}$$

where, $E_e$ is the irradiance, $\lambda$ is the wavelength of light used (470 nm), and $w_{loss}$ is the scaling factor for loss of photons. We used $g_{ChR2} = 0.17$ mS/cm$^2$ [22], and calculated $S_{cell}$ as area of a cylindrical cell of length = 100 $\mu$m and diameter = 16 $\mu$m. For a detailed description of other parameters and their values, we refer the reader to [36]. We found that for human atrial tissue, $E_e < 0.095$ mW/mm$^2$ were typically sub-threshold, meaning, they failed to trigger action potentials, whereas, higher light intensities successfully stimulated cells and initiated waves in extended media. In neonatal mouse, $E_e < 0.03$ mW/mm$^2$ was considered sub-threshold. The source code for our numerical simulations is provided in S1 Code.

## Results

When visible light is applied to the surface of the heart, its amplitude decreases exponentially with depth inside the heart wall [22]. Thus, a supra-threshold optical stimulus at the surface results in sub-threshold excitation beneath the surface. This implies that a sequence of uniformly timed optical stimuli most likely results in periodic sub-threshold perturbation of excitability inside the heart wall.

To study the effect of sub-threshold stimulation on the dynamics of a spiral wave in the heart, we start by considering the simplest scenario in which the arrhythmic heart is illuminated constantly and uniformly with light of sub-threshold intensity. We find that an increase in the applied light intensity (quantified as the irradiance, ($E_e$) leads to an increase in the core size of the meandering spiral, while preserving the general shape of the tip trajectory (Fig 1A, 1B and 1D). The dominant frequencies of rotation of the spiral decrease with increase in $E_e$ as

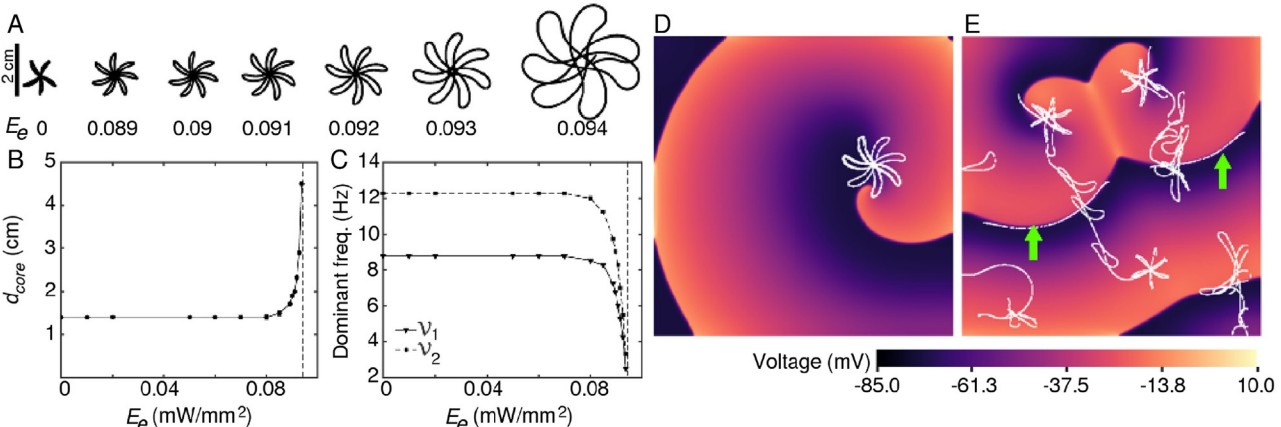

**Fig 1. Spiral wave dynamics in optogenetically-modified 2D human atrial tissue with uniform global sub-threshold illumination.** A. Tip trajectory of the spiral at different irradiance values ($E_e$). B. Increase in diameter ($d_{core}$) of the spiral core with increase in $E_e$. Here $d_{core}$ measures the diameter of the (outer) circle that just encloses the tip trajectory. C. Decrease in the dominant (first and second fundamental) frequencies of the spiral with increase in $E_e$. D. Trajectory of the spiral wave at constant, uniform, global illumination at $E_e = 0.092$ mW/mm$^2$. E. Break up of the spiral wave at the same $E_e$, when the light is applied as a periodic perturbation at a frequency of 1.0Hz. Green arrows point to the lines of the conduction block (shown in bold white). In B and C, the dashed line at $E_e = 0.095$ mW/mm$^2$ marks the limit of sub-threshold stimulation. In (A) and (D), the light was applied for 5 s.

illustrated in Fig 1C. Note that, both core size and dominant frequency of the spiral wave remain unaffected between $E_e = 0.0$ and $0.085$ mW/mm$^2$, beyond which value, these parameters exhibited sharp increase or decrease until $E_e = 0.095$ mW/mm$^2$, which marks the threshold of optical stimulation.

However, if the light is subsequently switched off, or a periodic illumination is applied with alternate light and dark phases, interesting dynamical effects emerge. In particular, uniform, global, time-periodic light pulses of $0.085$ mW/mm$^2 \leq E_e < 0.095$ mW/mm$^2$ led to the initiation of wavebreaks, as shown in Fig 1E. These breaks occur during the illuminated phase or at the beginning of the 'dark' phase of the applied stimulus, i.e., when the light was turned off in a pacing cycle. Application of discrete high sub-threshold perturbations to membrane voltage led to the spontaneous emergence of conduction blocks within the domain (see bold white lines indicated by green arrows in Figs 1E, 2B and 2C), soon after light is switched on. These blocks promote wave break initiation. For the full sequence of events leading to the incidence of wavebreaks, see Fig 2 and S1 Video. Voltage time series data from five representative points (128,128), (128,384), (384,384), (384,128) and (256,256) within the 512 x 512 domain are recorded for 5s of simulation. The data shows a large increase in APD and a drop in the period of the electrical activity, during illumination.

Thus, to investigate the cause for the appearance of wave blocks in a system that is known to promote supernormal velocities [21], we turned our attention to the $E_e$-dependence of the dispersion of APD and CV, i.e. the restitution properties of the system. Our results, as shown in Fig 3A and 3B, pointed to a strange anomaly: Both the APD and CV restitution curves for different irradiance values appeared to have gentle slopes (slope = 0.05, 0.07 and 0.071, respectively, for $E_e = 0.085$ mW/mm$^2$, 0.09 mW/mm$^2$ and 0.092 mW/mm$^2$) without spatial dispersion, which is fascinating because this type of wave break intiation is typically associated with steep restitution curves (slope $> 1.0$), or dispersion in restitution properties within a system (contrary to our case), or in the presence of local ionic heterogeneities (as in case of ischemia) [37, 38] or in domains with progressively degrading excitability, which results in the gradual flatenning of the CV restitution curve with reduced mean CV, characteristic of type-II ventricular fibrillation.

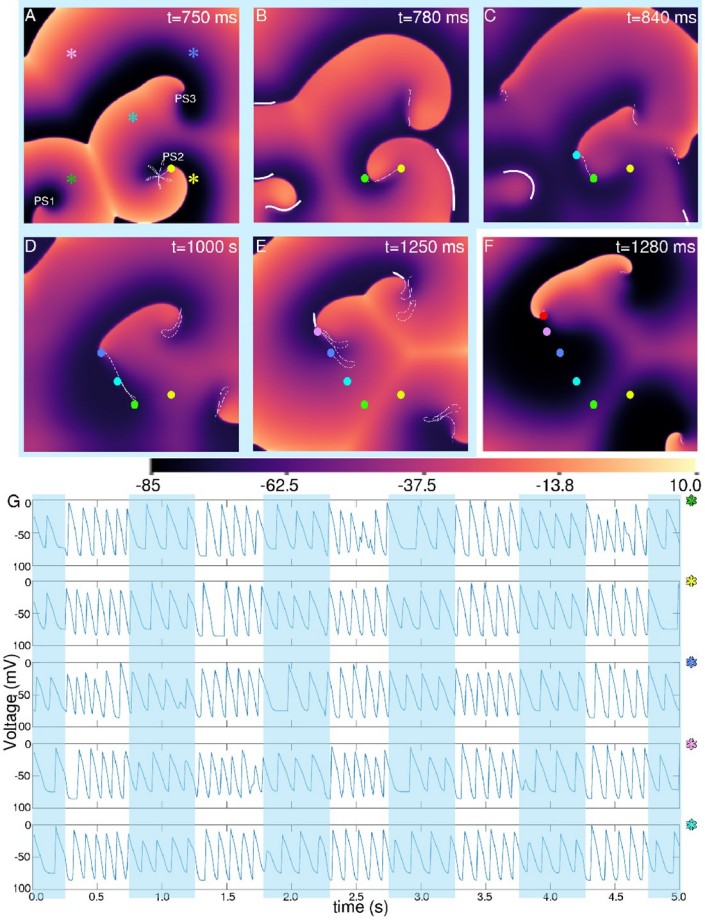

**Fig 2. Demonstration of spiral breakup with pulsed, uniform, global subthreshold illumination, for a light pulse applied from 750 ms, up to 1250 ms.** A-F. Spatial distribution of membrane voltage at different time points during (A-E, blue shaded) and after (F, non-shaded) one illumination cycle. The dashed white lines indicate the trajectories of the phase singularities (PS1, PS2 and PS3, shown in A), as they evolve since the preceding time point. Here, we highlight the spatiotemporal evolution of PS2. The hypocycloidal pattern of evolution of PS2 represents the dynamics of PS2 prior to the application of light. The filled yellow, green, cyan, blue, violet and red circles in A-F indicate the positions of PS2 at 750 ms, 780 ms, 840 ms, 1000 s, 1260 s and 1280 s, respectively. The bold white lines mark the zones of conduction block at these particular time points. Absence of a dashed white line in F, together with a sharp increase in the repolarized area, relative to E, show that after removal of light in E the activity remains frozen for a brief period of time (here, 30 ms). This allows the excitable tissue to recover. G. Voltage time series recorded from five representative points within the domain, as marked on A with the pink, green, yellow, blue and cyan asterisk symbols.

In addition, constant application of light to the domain resulted in supernormal CVs, which was in direct conflict with our observation of the emergence of lines of propagation block, just prior to the occurrence of the break.

To further investigate the origins of the conduction blocks in our 2D system, we tried to reproduce the block in pseudo-1D, using a rectangular simulation domain containing 512×10 grid points. We applied high frequency electrical stimulation to the left edge of the domain (Fig 3C), while illuminating the entire domain uniformly, in a time-periodic manner. In the absence of illumination, each applied stimulus resulted in the initiation of a new wave that propagated uninterruptedly to the right edge of the domain (Fig 3C).

However, in the presence of an optical stimulation with $E_e \geq 0.09$ mW/mm$^2$ and low frequency, i.e., $< 2.0$ Hz we observed a random blocking of the stimulated waves, always at the

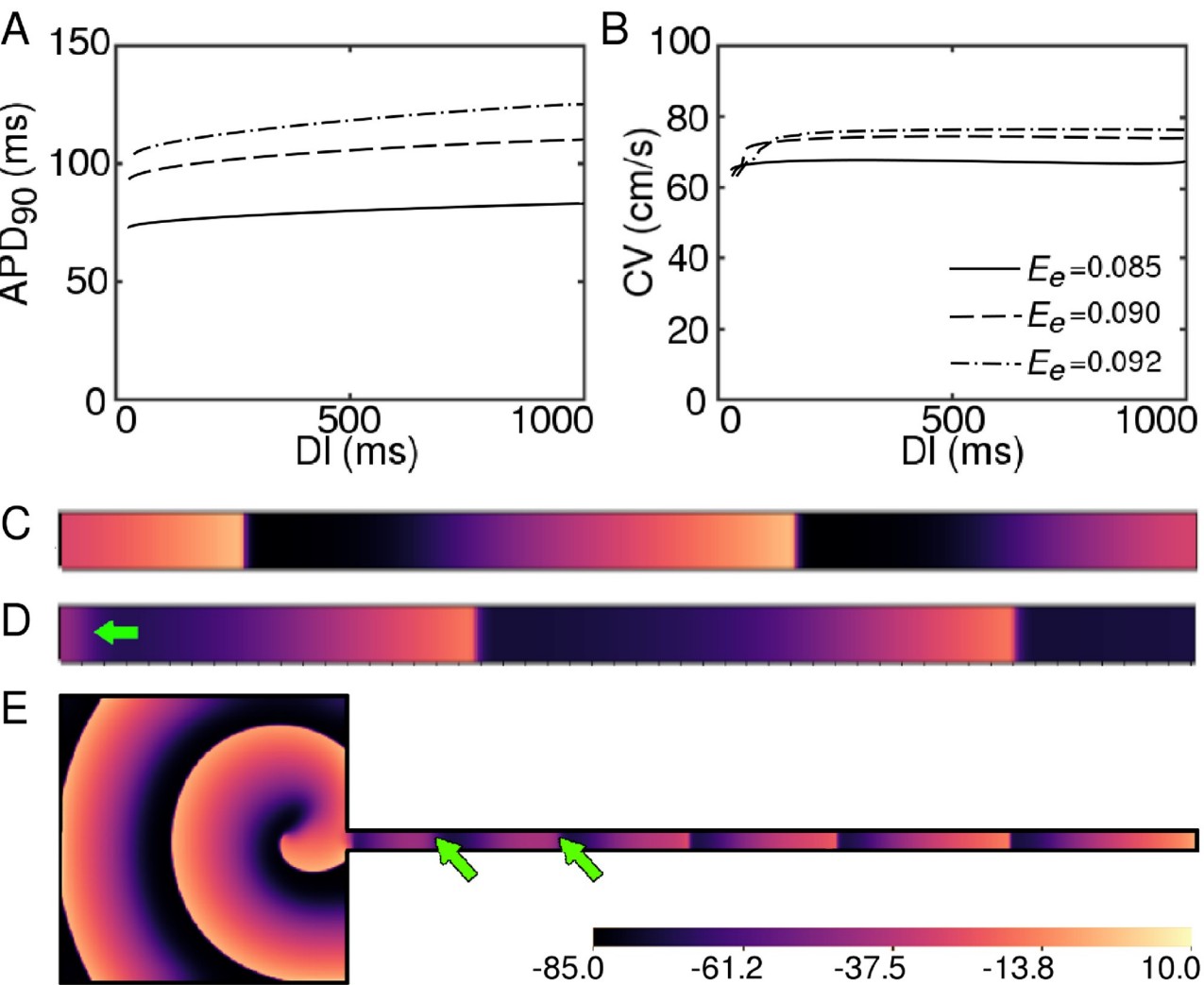

**Fig 3.** A. APD$_{90}$ and B CV restitution curves at $E_e$ = 0.085 mW/mm$^2$ (solid), $E_e$ = 0.09 mW/mm$^2$ (dashed) and $E_e$ = 0.092 mW/mm$^2$ (dot-dashed), for constant global illumination. C. Right propagating plane waves in a non-illuminated 512 × 10 pseudo-1D domain, paced electrically from the left boundary at 11.5 Hz. D. Wave block at the pacing site when the simulation in (C) is conducted in the presence of a uniform global time-periodic illumination (1.0 Hz, $E_e$ = 0.092 mW/mm$^2$). E. Wave block (indicated by green arrows) away from the influence of the spiral in 2D.

site of stimulation (Fig 3D). Since waveblock at the site of stimulation can occur due to various factors such as loss of capture, effect of CV restitution, interaction between new wavefront and slow waveback of the previous wave etc., our results remained inconclusive. To obtain wave-blocks inside the domain, away from the site of stimulation, we shifted the region for optical perturbation to 98 < x < 512, and resumed periodic electrical pacing at 11.5Hz (dominant frequency of the spiral wave in the absence of illumination), from the left end (x < 4) of the stripe. Our studies showed that every stimulus was capable of producing a new wave that propagated all the way to the right end of the stripe without interruption, although 1D traces along the length of the stripe did give us signatures of the effect of sub-threshold stimulation on the membrane potential in the illuminated region. Thus, clearly, the 1D reduction of our 2D system was sub-optimal for wave block visualisation.

The key difference, in our view, was the applied stimulus itself, which happened to comprise two fundamental frequencies of the spiral in 2D, but only one frequency in 1D. To overcome

this challenge, we designed a hybrid domain by connecting our original 2D simulation domain ($512 \times 512$ grid points) to a pseudo-1D stripe ($1536 \times 30$ grid points) and used the spiral in the 2D part to drive the electrical activity in the stripe. Since there were no sources of excitation within the stripe region, this geometry had no bearing on the dynamics of the spiral itself. We then applied uniform global optical stimulation at $E_e = 0.092$ mW/mm$^2$ and frequency 1.0Hz to the stripe (pseudo-1D) region, without affecting the 2D domain. This setup allowed us to observe the formation of conduction blocks well within the pseudo-1D region, away from the spiral. This observation could be reproduced for a wide range of frequencies of the applied light stimulation (Fig 3E, and S2 Video). In addition to the disappearance of some of the waves that had propagated into the pseudo-1D domain over a certain distance, we observed spatio-temporal oscillation in the wavelength of successive waves.

To understand the basis for these oscillations and correlate them with the occurrence of conduction block, we examined the effects of sub-threshold stimulation in a single-cell mathematical model of the human atria, under the influence of light applied to different phases of an electrically evoked AP (Fig 4). Our studies demonstrate a phase-dependent response of the *APD*. The actual value of the *APD* increases or decreases by an amount that is determined by the level of repolarization of the membrane at the beginning of the stimulation (see Fig 4A–4F). In our case, the applied light stimulus caused maximum increase in $APD_{90}$ when it was applied at a membrane repolarization level of $\approx 64\%$, i.e., at the excitation threshold of the cardiac cell. Further repolarization of the membrane caused $APD_{90}$ to decrease below its unperturbed value, reaching a minimum ($\approx 2\%$ decrease) when the cell membrane had repolarized by $\approx 90\%$, i.e., at the start of the refractory period in an unperturbed cell. (Fig 4G).

A direct consequence of this differential response is a phase-dependent delay of recovery, introduced along the length of a wave, in extended media. Such a delay leads to differences in CV along a wavelength. In the presence of light, the wavefront propagates at supernormal speeds while the waveback slows down. This results in (*i*) prolongation of the wavelength with a gradual change in the shape of wave profile and (*ii*) blocking of the following wave by the preceding one. When light is applied to a propagating wave, cells located at increasing distances from the wavefront towards the waveback, receive light at progressively shifted phases of their AP (Fig 4A–4E). This results in a gradual decrease in the rate of recovery of the cells in between the wavefront and the waveback, thereby changing spatial profile of the wave, to give it a pinched appearance. (Fig 5A–5C). Instead of the constant, uniform, global illumination, if the light is now applied periodically in time, the pinched wavelength gets the opportunity to return to its original state during the light-off period (Fig 5D–5F). This explains the occurrence of spatiotemporal oscillations in the wavelength of the propagating wave (in 1D) or a spiral arm (in 2D).

The higher the $E_e$, the stronger the conditioning of the wavelength, i.e. the wave appears more pinched. Immediately after the light is switched off, the differences between the recovery rates of the cells at different distances from the wavefront disappear.

The cells on the wavefront that allowed propagation at supernormal CV recover slowly, allowing the wavefront to expand from its pinched state in a gradual manner. The cells on the wave back that were forcibly held at a sub-threshold voltage higher than the normal resting membrane potential (RMP) in the pinched state recover immediately, creating a window for re-excitation. In 2D, this window allows re-entry, as shown in Fig 6. The inset shows the spatial pattern of electrical activity in the domain, which takes up a spiral form. By tracing the voltage distribution along the solid white line (see Fig 6, inset) in the direction of the arrow, we illustrate how a right-propagating wave is modulated over time with an optical perturbation ($E_e = 0.092$ mW/mm$^2$, frequency 1.0 Hz) to create a vulnerable window (marked with a filled gray

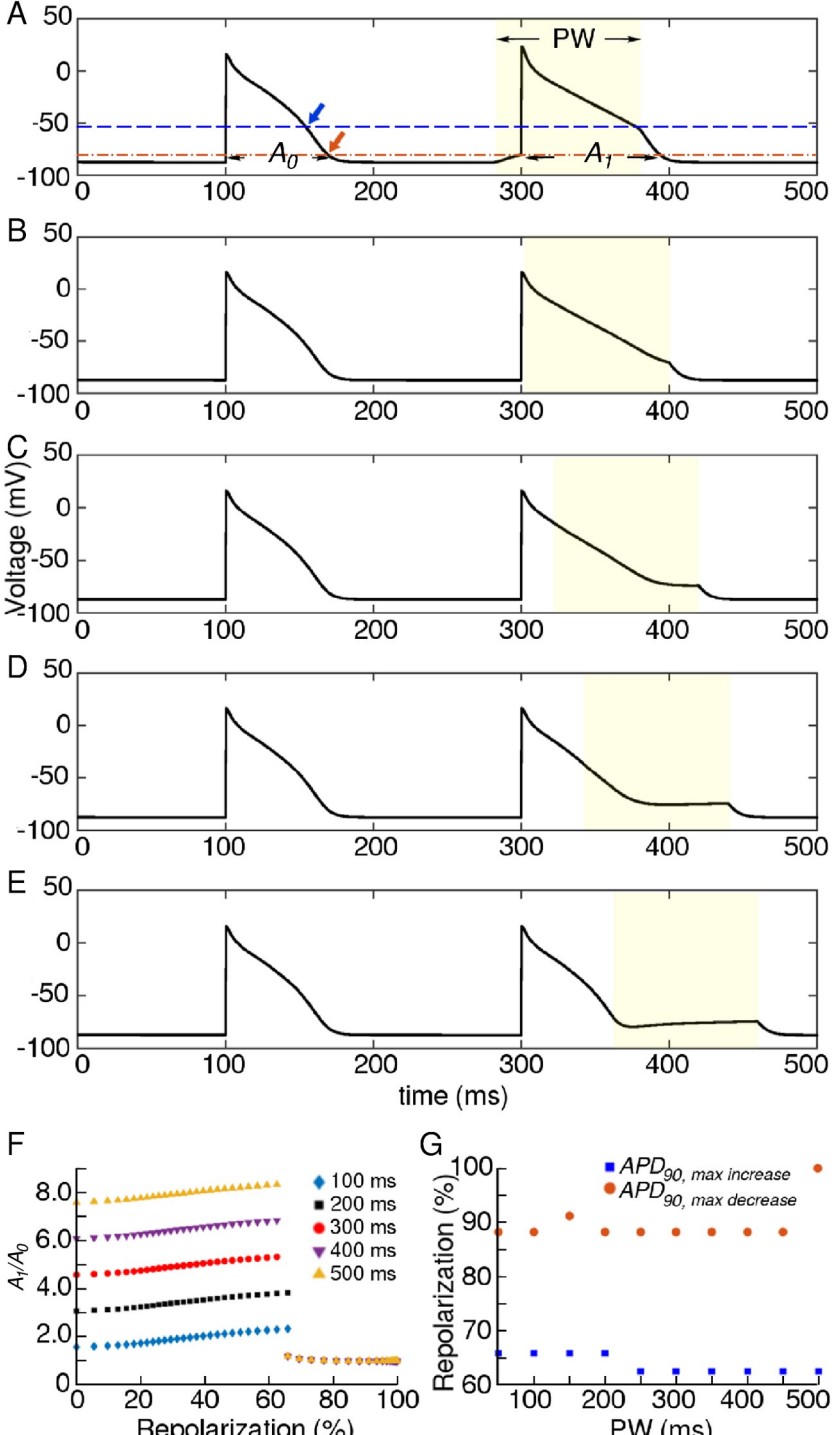

**Fig 4.** A-E Effect of applying a light pulse (filled yellow rectangle) (0.092 mW/mm$^2$, 100 ms) to different phases of an action potential (AP). $A_0$ and $A_1$ refer to the $APD_{90}$ values of the unperturbed and perturbed AP, respectively. F Dependency of $A_1/A_0$ on the percentage of repolarization of the cell membrane at the start of stimulation (for different PWs). G PW-dependence of the percentage repolarization, at the start of a stimulation that causes maximum increase (blue) or decrease (brown) in $APD_{90}$. The dashed blue and brown lines on A correlate the findings in (G) to real values of the membrane voltage. The blue and brown arrows in A indicate where the light stimulus should be applied in order to achieve maximum increase or decrease in $APD_{90}$, respectively.

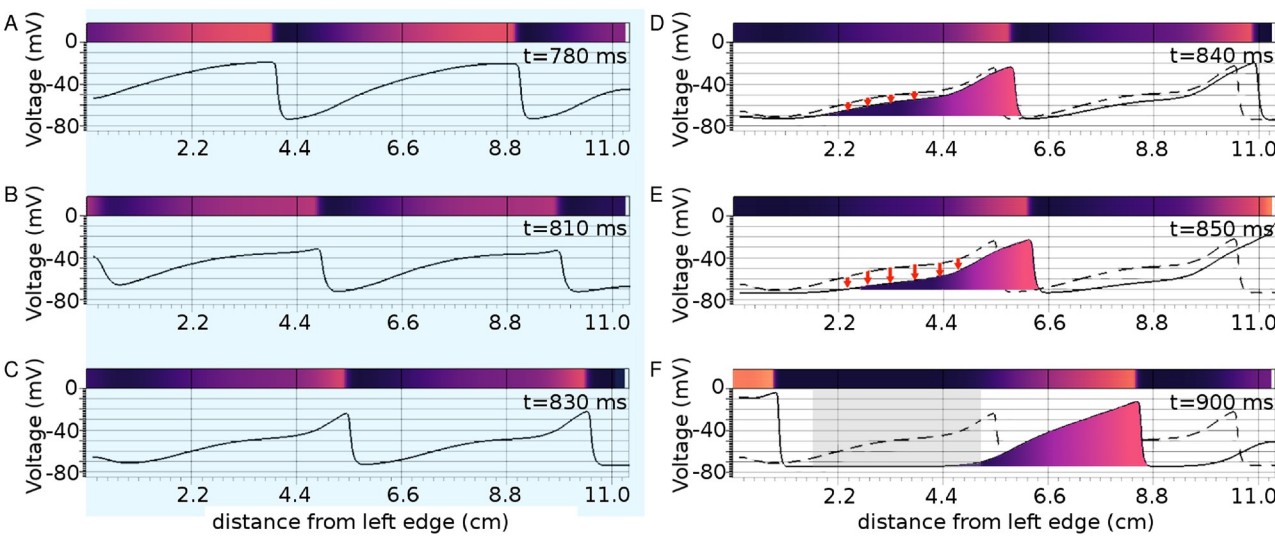

**Fig 5.** A-C Effect of applying a light pulse (0.092 mW/mm², 1000 ms) to a plane wave in a 512 × 10 pseudo-1D domain. D-F Rapid recovery of the wave tail (indicated by bold red arrows) along the line y = 5 of the domain, immediately after the light is turned off. The wave profile reshapes and widens the spatial window of vulnerability (gray rectangle in F). The colored panel in each sub-figure illustrates the wave profile as it propagates through the domain.

rectangle). The wave break occurs at $\approx$ 3900 ms, leading to re-excitation and reentry. This is marked by the reversal of wave direction.

Finally, to test the generic nature of this new mechanism of wave break initiation at stable parameter régimes, we repeated the study in another system, a 2D model of neonatal mouse ventricular tissue. At uniform global constant illumination with LI = 0.025 mW/mm², a spiral wave rotated with no signatures of breakup (Fig 7A). However, when stimulated periodically at 4 Hz, with light of the same intensity and pulse length 200 ms, breakup similar to Fig 1E was observed (Fig 7B). A study of the restitution characteristics for this system, as presented in Ref. [39] (see Figs 5A and 5B, therein), indicated that wavebreaks occurred in a stable parameter régime, thus proving the robustness and restitution-independence of our proposed mechanism for wavebreaks.

## Discussion

There are several theories regarding the factors and conditions that cause a spiral wave to break up. Of these, the strongest candidate is restitution (both APD and CV) in cardiac tissue [40]. Some studies argue that APD and CV restitution curves fundamentally characterize the wave dynamics in the heart as they reflect the mesoscopic effects of changes in ion currents and concentrations occurring at the cellular level [40].

Typically, a steep APD restitution curve (slope $\geq$ 1.0) can drive the system via Hopf bifurcation to APD oscillations [41–43]. Such alternans promotes the development of functional conduction blocks because the propagation of a "long" action potential (AP) into a "short" AP region fails, thereby limiting the allowable diastolic interval (DI) [40]. Another proposed mechanism, again relating to the role of a steep APD restitution curve, suggests that the steepness of the curve is negatively correlated with the speed of the waveback in the presence of recovery gradients [44]. A gradient $\geq$ 1.0 introduces a difference between the CVs of the propagating wavefront and back so that within a wave train the front of the following wave collides with the back of the preceding wave, resulting in a functional conduction block [42, 44–46].

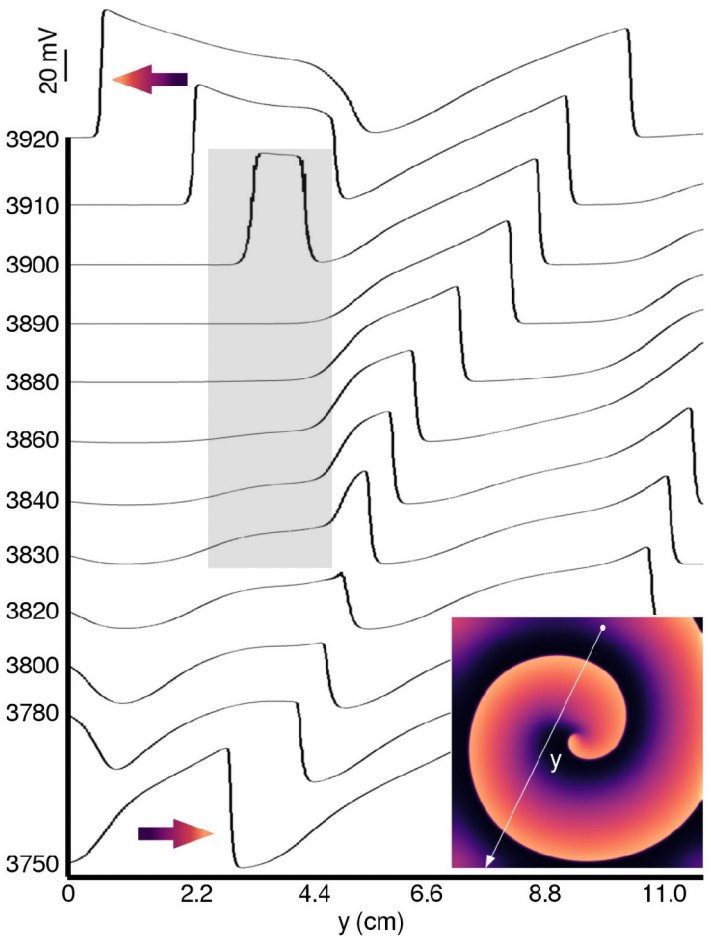

**Fig 6. Mechanism of spiral wave breakup in 2D.** The time-stamped 1D voltage traces measured along **y** (solid white arrow in the inset) show that in a globally illuminated domain ($E_e$ = 0.092 mW/mm$^2$, frequency = 1.0 Hz), CV of propagating wavefronts increases, whereas that of wavebacks decreases. This modulates the spatial profile of the wave. In our example, a vulnerable window appears between 3830 ms and 3900ms (indicated by filled gray rectangle). Re-excitation and reentry occur at 3900 ms.

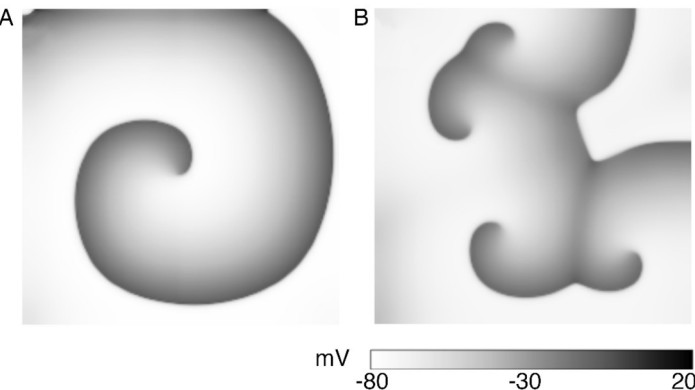

**Fig 7. Spiral wave breakup in a 2D model for optogenetically-modified neonatal mouse ventricular tissue.** A Intact spiral in the presence of uniform, global, constant illumination ($E_e$ = 0.025 mW/mm$^2$). B Break up in the presence of periodic optical stimulation (0.025 mW/mm$^2$, 4 Hz, PW = 200 ms).

Other mechanisms of wave break include the occurrence of a spatially non-synchronized type of APD oscillation, also known as spatially discordant APD alternans [47–50], the presence of a biphasic APD restitution curve [51, 52], effects of hysteresis [53–56] and Doppler shift through the trajectories of the spiral peaks [57].

A fascinating study by Fenton et al. [58] on wave break initiation in ischemic heart tissue shows that dynamic flatenning of the APD restitution curve (characterized by slope < 1.0), results in an electrical 'memory' effect, which provokes a monotonic reduction in the excitability of cardiac tissue. Over time, the excitability becomes so low that a spiral breaks up close to its tip in the form of a 'voltage drop'. The study concludes that spiral break-up can be induced in parameter régimes with flat APD restitution as long as the dynamic effects of tissue memory and substantially large changes in basic cycle length are taken into account. Our study is consistent with the work of Fenton et al. in that wavebreaks are observed at low APD and CV restitution slopes. However, we show that for a system with static APD and CV restitution curves, in the absence of a factor leading to a progressive reduction in tissue excitability and without explicit inclusion of the electrical memory effect, it is still possible to initiate wavebreaks that develop into sustained re-entry.

Another study by Zemlin et al. [59] reports spiral wave break-up in parameter regimes characterized by strongly negative slopes of the APD restitution curve. However, as noted in their work, such break-up is often less pronounced than the break-up that occurs with positive slopes, mainly because break-up with negative restitution takes a fairly long time to mature, compared to break-up with positive restitution. The wave break initiation that we demonstrate in this article happens very fast, in most cases, requiring not more than 1–2 pulses.

In experiments performed on the rabbit heart, Banville and Gray [60] demonstrates the crucial role of the spatial distribution of APD- and CV restitution on the onset of alternans and fibrillation. In contrast, we use a homogeneous domain (with all cells identical and subjected to the same external conditions) that even in the absence of a spatial dispersion in APD and CV restitution, wave break initiation can occur. This leads one to reconsider the underlying factors involved in wave break initiation in cardiac tissue.

A rare, but quite efficient, mechanism of spiral wave break up involves supernormal excitability in certain domains. It is associated with a decrease in the excitation threshold with decreasing diastolic interval (DI). Typically, if it exists in a model, it promotes the development of conduction blocks through collisions between the rapidly advancing wavefronts and subsequent stacking at short DI, eventually leading to the initiation of wavebreaks. Breakup occurs particularly close to the spiral tip. Supernormal CV can also lead to heterogeneity of refractoriness at shallow APD restitution, in which case scalloping occurs [40]. The mechanism of spiral wave break initiation that we report in the present article is somewhat similar to this phenomenon. With uniform global light perturbations, we are able to impose supernormal CVs on the propagating electrical activity. This results in the development of wave blocks within the domain that are visible as white lines in Fig 1E. In our studies, the reentry occurs once the factor causing supernormality is withdrawn, i.e., the light perturbation is removed, allowing the wavefront and wave back to restore their normal velocities.

Here, we attribute the onset of the mechanism of wavebreak initiation to the elevation in the resting membrane potential upon illumination. Elevation of the resting potential results in the prolongation of the recovery time of the Na channel. In the model, it is described by the time constant of the j-gate. This influences the CV restitution at short diastolic intervals and prolongs the effective refractory period to potentiate conduction block [61, 62]. Application of light to the domain containing the spiral wave causes the wavefront to speed up, but the waveback simultaneously slows down. This leads to wavefront-waveback interactions which further contribute to the formation of conduction blocks. Our studies show that these conduction

blocks appear soon after the light is turned on. Subsequently, existing electrical activity in the doamin to propagate around the blocked regions, resulting in the development of wavebreaks. However, spiral waves formed around these wavebreaks rotate with much longer periods and larger cores, as compared to spirals formed in the absence of illumination. Thus in many cases, these spirals survive only for a few cycles. On the contrary, when a wavebreak is formed by switching off the light (as demonstrated in Fig 2F), the electrical activity within the domain freezes temporarily, allowing the excitable tissue around the break to recover. The spatial distribution of repolarized tissue re-organizes and a spiral is formed with shorter period and smaller core. Thus wavebreaks triggered at the start of the 'dark' period lead to sustained reentry.

Finally, the effect of global sub-threshold stimulation was studied in simple ionic models of cardiac tissue, in the context of electrical defibrillation. A particularly close phenomenon was reported by Sridhar and Sinha [63], who demonstrated the possibility to terminate electrical activity using a time-invariant global sub-threshold electrical stimulation to achieve synchronization. Their study indicated that an electric pulse of length $\mathcal{O}(1APD)$ was sufficient to cause this effect. Our studies showed (see S3 Video) that if the stimulus in [63] was instead, applied at low frequency (1–2Hz), using pulses of very short duration ($\mathcal{O}(0.1APD)$), then such perturbation could also initiate wavebreaks. However, such breaks occur within a very restricted parameter régime and critically rely on the frequency of the applied perturbation for sustenance, owing to the sensitive dependence of the response of the single cell AP morphology, on the phase of the AP at which the stimulus is applied.

Our study provides a clear example of a mechanism of wave break initiation and reentry that occurs when the tissue does not exhibit any of the standard markers of excitable medium vulnerability, i.e. slope of the APD restitution curve is less than 1.0, the CV restitution curve is relatively flat, and there is no factor causing dispersion of restitution properties within the domain. As we demonstrate using two different animal models, the phenomenon is generic and does not depend on the selection of model used. The only reason, we used a model of chronic AF remodelling, is because it makes more sense to 'defibrillate' a diseased substrate rather than the healthy one, where fibrillation is less likely to occur. The proposed mechanism serves as a candidate that has the potential to cause failure of low-amplitude defibrillation using pulsed global (electrical or optical) stimulation. We further believe that this mechanism is important in the context of optogenetic defibrillation, which is a long-term vision for many cardiac researchers, because of its promise for painless defibrillation. Since the human atrial wall is very thin, many researchers find it technically most optimal for optogenetic defibrillation. However, any step towards translation of this technique, necessitates the development of an in-depth understanding of not only the mechanisms that control, but also initiate, spiral waves. This holds true for low-energy defibrillation in general, where weak electric pulses, which are used to terminate spiral waves, can also potentially initiate them.

## Conclusion

In addition to the general notion that suprathreshold perturbations have a profound effect on electrical wave dynamics in cardiac tissue, we demonstrate a situation in which small-amplitude electrical perturbations can lead to wave break initiation and reentry in tissue that otherwise does not exhibit any of the standard markers of excitable medium vulnerability. In this work, the applied subthreshold perturbations of membrane voltage lead to the onset of dynamic instabilities. These instabilities differ from the classical "Early After Depolarization" (EAD) type instabilities that are known to contribute to spatiotemporal chaos in cardiac tissue [64]. Unlike classical spatiotemporal instabilities, which either exist within the system or arise

naturally, the instabilities reported in this study are externally induced and occur only in response to discrete perturbations. Consequently, the system exhibits a stable state characterised by many spiral waves. However, an electrical signal recorded from any part of the domain does not show a broadband frequency spectrum characteristic of classical electrical turbulence. The discovered mechanism has the potential to inhibit low-amplitude defibrillation using pulsed global (electrical or optical) stimulation. In the context of optogenetics, this observation is of crucial value as it provides useful insights into the translational prospects of optogenetic defibrillation in the human atria. The key to success lies in understanding and exploiting the mechanisms of creation and annihilation of spiral waves. Thus, our study makes an important contribution to the advancement of the field of cardiac optogenetics and arrhythmias, in general.

## Supporting information

**S1 Code. The source code used for obtaining the results that are presented in this paper.**
(GZ)

**S1 Video. Controlled break-up of spiral waves, induced by periodic sub-threshold optical stimulation, at frequency 1.0 Hz.** The trajectory of the spiral tip(s) is traced using white lines that fade over time. The filled blue circle below the voltage color-code (in mV) in select frames indicates the application of uniform, global sub-threshold illumination at $E_e = 0.092$ mW/mm$^2$, in these frames.
(MOV)

**S2 Video. Reduction of the 2D system to a hybrid 1D system, in which we combine a 2D simulation domain to a pseudo-1D outlet.** The electrical stimulation in the pseudo-1D domain is driven by the unperturbed spiral wave in 2D. A uniform global periodic sub-threshold stimulation (frequency: 1.0 Hz) with light of intensity 0.092 mW/mm$^2$ is applied to the pseudo-1D domain. Propagation failure occurs well within the pseudo-1D domain for randomly selected waves. In addition, we observe oscillations in wavelength and modulation of the wave profile for waves propagating through the pseudo-1D domain.
(MOV)

**S3 Video. Break-up of spiral waves, induced by global periodic sub-threshold electrical stimulation, at frequency 1.0 Hz, with a pulse length of 20 ms.** The filled red circle appearing at the top right corner (of select frames) of the video indicates the application of the stimulus in these frames.
(AVI)

## Acknowledgments

This work was supported by the Max Planck Society for providing computational resources.

## Author Contributions

**Conceptualization:** Rupamanjari Majumder, Vladimir S. Zykov, Eberhard Bodenschatz.

**Data curation:** Rupamanjari Majumder.

**Formal analysis:** Rupamanjari Majumder, Sayedeh Hussaini.

**Investigation:** Rupamanjari Majumder, Sayedeh Hussaini, Vladimir S. Zykov, Eberhard Bodenschatz.

**Methodology:** Rupamanjari Majumder.

**Project administration:** Rupamanjari Majumder, Eberhard Bodenschatz.

**Resources:** Rupamanjari Majumder, Stefan Luther, Eberhard Bodenschatz.

**Software:** Rupamanjari Majumder, Sayedeh Hussaini.

**Supervision:** Rupamanjari Majumder, Vladimir S. Zykov, Eberhard Bodenschatz.

**Validation:** Rupamanjari Majumder, Sayedeh Hussaini, Stefan Luther.

**Visualization:** Rupamanjari Majumder, Sayedeh Hussaini.

**Writing – original draft:** Rupamanjari Majumder, Sayedeh Hussaini, Vladimir S. Zykov, Stefan Luther, Eberhard Bodenschatz.

**Writing – review & editing:** Rupamanjari Majumder, Sayedeh Hussaini, Vladimir S. Zykov, Stefan Luther, Eberhard Bodenschatz.

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
