## [Decision Letter · Decision Letter 0]

25 Aug 2021

Dear Dr Majumder,

Thank you very much for submitting your manuscript "Pulsed low-energy stimulation initiates electric turbulence in cardiac tissue" for consideration at PLOS Computational Biology.

As with all papers reviewed by the journal, your manuscript was reviewed by members of the editorial board and by several independent reviewers. In light of the reviews (below this email), we would like to invite the resubmission of a significantly-revised version that takes into account the reviewers' comments.

We cannot make any decision about publication until we have seen the revised manuscript and your response to the reviewers' comments. Your revised manuscript is also likely to be sent to reviewers for further evaluation.

Sincerely,

Alison Marsden

Associate Editor

PLOS Computational Biology

Daniel Beard

Deputy Editor

PLOS Computational Biology

Reviewer's Responses to Questions

**Comments to the Authors:**

Reviewer #1: In this study, Majumder et al carried out computer simulations of atrial tissue models to investigate the effects of optogenetically induced subthreshold current stimuli on spiral wave dynamics. They showed that with a constant subthreshold current, the spiral tip meandered with a larger and larger core region as the current strength increases, and no breakup was observed. But when the current with a certain strength was switched on and off, wavebreak occurred. They carried out additional simulations to investigate the mechanism of wavebreak induced by the on-and-off of the subthreshold current stimuli.

1) The study provides new insights on the role of optogenetics in cardiac conduction dynamics, however, the mechanism of wavebreak was not very clearly demonstrated or explained (see comments below).

2) The authors were trying to explain the mechanism of wavebreaks by APD (and APD restitution) and CV (and CV restitution) as for spiral wave breakup investigated in many previous studies. Although they mentioned the change of excitability during the switch-on and switch-off of the light, I do not think that they explained correctly the “true” mechanism or causes of wavebreak. One major concern is the definition of APD in Fig.3, which is somewhat ambiguous, even incorrect. First, although APD90 was used, I was not able to find a clear definition of APD in the manuscript. Second, if one uses APD80, APD70, …, then the APD responses to the optogenetic current will be drastically different from what is shown in Fig. 3F. Therefore, which APD is a proper one to use to explain the observed wave dynamics is a problem.

3) I do not think that the mechanism of wavebreak is as complex as explained by the authors. It is not related to APD or APD restitution, but to the change of refractoriness or excitability due to the elevation of the resting potential caused by the optogenetic current. As shown in Figs.3 D and E, the effect of the current in the diastolic phase is to elevate the resting potential. It is well known that elevation of resting potential first increases CV (this is what occurred in Fig.2B for large DI) and then decreases CV due to the competition between Na channel inactivation and the threshold of excitable, i.e., a high resting potential causes more Na channel inactivation but it is closer to the threshold of Na channel activation. A typical example of this is acute ischemia. The spiral wave behavior in response to ischemia (see Figs.1 and 2 in Xie et al, Am J Physiol. 280, H1167(2001)) is very similar to what are shown in Figs.1 A and B in the present study. Another effect of elevation of the resting potential is the prolongation of the recovery time of the Na channel. In the model, it is described by the time constant of the j-gate. This will change the CV restitution (this is why CV decreases faster as DI decreases at very short DIs in Fig.2B) and prolong the effective refractory period to potentiate conduction block. This effect was demonstrated by Xie et al [Heart Rhythm 6, 1641(2009), Fig.7 and Fig.S4] in a study of fibroblast-myocyte coupling in which the resting potential was elevated due to the fibroblast-myocyte coupling. Conduction block caused by elevation of the resting potential was also studied by Liu et al (Heart Rhythm 12, 2115 (2015)) on the role of subthreshold DAD on conduction block. Based on the above studies, I believe that the local conduction block or wavebreak is caused by the elevation of the resting potential in a short time after the onset of the optogenetic stimulus current due to the sudden prolongation of the refractory period.

4) In the manuscript, the authors discuss spiral wave breakup and tried to make some link of the wavebreak in the current study to those in the previous studies. Note that the spiral wave breakup occurs as spatiotemporal instabilities (see a review by Qu et al, Phys Rep 543, 61(2014)), which is not the case in the present study. As shown in Fig.1A by the authors, no breakup occurs before the current becomes suprathreshold. Wavebreaks only occur when the current is periodically switched on and off, indicating that wavebreak is caused by a sudden change of the current/parameters with the mechanism explained above.

5) The last sentence in the abstract “This finding changes the paradigm of cardiac arrhythmia research …” is overly stated. It provides some useful insights on optogenetics but not shifts the paradigm of arrhythmia research.

6) The sentence in the abstract “This mechanism involves ‘conditioning’ or reshaping the wave profile from front to back, such that, removal of the external light source causes rapid recovery of cells at the waveback, leading to the emergence of vulnerable windows for sustained re-entry in spatially extended systems.” seems to imply that conduction block occurs at the moment of removal of the external light. However, by watching the movie, I saw that conduction block always occurred at the onset of the external light (the moment when the blue circle was on). During the on-phase or the off-phase, the spiral waves are stable (no breakup).

7) Fig.1E should be an expanded figure (as Fig.2) independent from Fig.1. I would suggest to plot voltage snapshots at different time points before, during, and after the switch on and off of the light. It would be also useful to show the period of the spiral wave (from one or two locations) versus time combined with light on and off. I understand that there is an online movie to show the wave dynamics, since the whole paper is about Fig.1E, it is important to show a clear picture or dynamics for the readers. It is not clear to me where and when the wavebreaks occur by reading text only, and it took me sometime by watching the movie to find out that the wavebreaks only occur right after the onset of the light (I hope that I’m correct on this).

8) It seems there is a Conclusion section, but the content is missing.

**Have the authors made all data and (if applicable) computational code underlying the findings in their manuscript fully available?**

Reviewer #1: **No: **not stated in manuscript where the codes are available.

PLOS authors have the option to publish the peer review history of their article (what does this mean?). If published, this will include your full peer review and any attached files.

Reviewer #1: No
---

## [Decision Letter · Decision Letter 1]

23 Sep 2021

Dear Dr Majumder,

We are pleased to inform you that your manuscript 'Pulsed low-energy stimulation initiates electric turbulence in cardiac tissue' has been provisionally accepted for publication in PLOS Computational Biology.

Best regards,

Alison L. Marsden

Associate Editor

PLOS Computational Biology

Daniel Beard

Deputy Editor

PLOS Computational Biology

Reviewer's Responses to Questions

**Comments to the Authors:**

Reviewer #1: I have one minor comment:

Line 286: "Typically, a steep APD restitution curve (slope > 1:0) can drive the system

via Hopf bifurcation to APD oscillations [42{44]." The APD restitution slope induced alternans should be a period-doubling bifurcation (like the first bifurcation in the Logistic map) or a pitchfork bifurcation, not a Hopf bifurcation in a paced single cell or even periodically paced tissue. Hopf bifurcation occurs in a ring (refs. 43 and 44) is because the presence of CV restitution and the periodic boundary condition (because it is a ring). In open boundary conditions, it results in spatially discordant alternans, I'd suggest to remove Hopf bifurcation, and simply state that a steep APD restitution promotes APD alternans and oscillations.

**Have the authors made all data and (if applicable) computational code underlying the findings in their manuscript fully available?**

Reviewer #1: Yes

PLOS authors have the option to publish the peer review history of their article (what does this mean?). If published, this will include your full peer review and any attached files.

Reviewer #1: No

---

## [Editor Report · Acceptance letter]

4 Oct 2021

PCOMPBIOL-D-21-01123R1 

Pulsed low-energy stimulation initiates electric turbulence in cardiac tissue

Dear Dr Majumder,

I am pleased to inform you that your manuscript has been formally accepted for publication in PLOS Computational Biology. Your manuscript is now with our production department and you will be notified of the publication date in due course.

With kind regards,

Anita Estes
